# Association between the Erythrocyte Membrane Fatty Acid Profile and Cognitive Function in the Overweight and Obese Population Aged from 45 to 75 Years Old

**DOI:** 10.3390/nu14040914

**Published:** 2022-02-21

**Authors:** Jingyi Shen, Jinchen Li, Yinan Hua, Bingjie Ding, Cui Zhou, Huiyan Yu, Rong Xiao, Weiwei Ma

**Affiliations:** 1School of Public Health, Beijing Key Laboratory of Environmental Toxicology, Capital Medical University, Beijing 100069, China; 13051118875@163.com (J.S.); jinchen_lee@163.com (J.L.); huayinan@ccmu.edu.cn (Y.H.); shian01zc@126.com (C.Z.); bjyuhuiyan@126.com (H.Y.); xiaor22@ccmu.edu.cn (R.X.); 2Department of Clinical Nutrition, Beijing Friendship Hospital, Capital Medical University, Beijing 100050, China; dbj8202@126.com

**Keywords:** overweight, obese, cognitive impairment, dietary fatty acid intake, erythrocyte membrane fatty acid profile

## Abstract

Dietary fatty acid intake is closely related to the cognitive function of the overweight and obese population. However, few studies have specified the correlation between exact fatty acids and cognitive functions in different body mass index (BMI) groups. We aimed to explain these relationships and reference guiding principles for the fatty acid intake of the overweight and obese population. Normal weight, overweight, and obese participants were recruited to receive a cognitive function assessment and dietary survey, dietary fatty acids intake was calculated, and the erythrocyte membrane fatty acid profile was tested by performing a gas chromatography analysis. The percentages of saturated fatty acids (SFAs) in the obese group were higher, while monounsaturated fatty acids (MUFAs) and polyunsaturated fatty acids (PUFAs) were lower than in the normal weight and overweight groups. In the erythrocyte membrane, the increase of n-3 PUFAs was accompanied by cognitive decline in the overweight group, which could be a protective factor for cognitive function in the obese group. High n-6 PUFAs intake could exacerbate the cognitive decline in the obese population. Dietary fatty acid intake had different effects on the cognitive function of overweight and obese people, especially the protective effect of n-3 PUFAs; more precise dietary advice is needed to prevent cognitive impairment.

## 1. Introduction

Obesity, which is defined as abnormal or excessive fat accumulation that may impair health [1], is known as an important risk factor for mild cognitive impairment (MCI), Alzheimer’s disease (AD), and dementia [2,3]. In the past few years, the prevalence of obesity has increased substantially. In total, 39% of adults aged 18 years and over were overweight in 2016, and 13% were obese worldwide [4]. According to the Chinese Nutrition and Chronic Disease Status Report in 2020, more than 50% of adults were overweight or obese [5]. A cross-sectional study from eight regional centres in REACTION (China’s Risk Evaluation of cAncers in Chinese diabeTic Individuals, a lONgitudinal study) found that more than 64.4% residents aged over 40 years were peripheral obesity and/or central obesity [6]. In 2020, China had 249.49 million people aged 60 years or older, representing 17.9% of the total population of 1.40 billion people [7], which led to the higher prevalence of MCI, AD, or dementia [8]. Cognitive impairment in the overweight and obese population, with its increasing prevalence, is becoming a crucial public health concern.

Dietary factors, especially high fat intake [9], could impact hippocampal neurogenesis and cognition, and could play an important role in the occurrence of obesity and cognitive impairment [10]. In a cross-section study, researchers revealed that a high percentage of fat in the diet was greatly associated with the increased risk of MCI [9]. As reported, saturated fatty acids (SFAs) and trans-fatty acids (TFAs) might have a negative correlation with cognitive functions, and monounsaturated fatty acids (MUFAs) and polyunsaturated fatty acids (PUFAs), particularly n-3 PUFAs, such as eicosapntemacnioc acid (EPA) and docosahexaenoic acid (DHA) [11], presented a protective effect against cognitive decline [12,13]. Epidemiological studies and animal experiments showed that MUFAs and PUFAs could play a role in reducing weight, and improving metabolic syndrome (MS) and cognitive impairment [14,15]. The obese population further exhibited decreased memory ability and executive ability, accompanied by abnormalities of brain structure [16], in comparison to those with normal weight. Taken together, dietary fatty acids might aggravate or alleviate the cognitive function of the overweight and obese populations to varying degrees.

In our previous study, the levels of SFAs in plasma were negatively associated with Montreal Cognitive Assessment (MoCA) scores, while MUFAs, n-3 PUFAs and n-6/n-3 PUFAs in plasma had a positive association with MoCA scores [17]. However, plasma fatty acid profiles could not reflect a long-term intake of fatty acids [18], which motivated us to use the fatty acid compositions in erythrocyte membrane as markers. Fatty acids are an integral part of cellular membranes [19], and the intake of fatty acids could modulate cell membrane fluidity [20]; in addition, erythrocytes have a circulatory half-life of 120 days in humans, such that the erythrocyte membrane fatty acid profile came to be considered to reflect the level of fatty acid intake in the recent few months [21,22]. Previous studies pointed out that lower PUFA and higher SFA proportions in the erythrocyte membrane fatty acid composition could perhaps predict cognitive impairment in the elderly in the Chinese population [23].

In this study, the association of dietary fatty acid intake with cognitive function in the overweight and obese population was assessed by using the erythrocyte membrane fatty acid profile. Our results indicated that the intake of fatty acids had a different influence on the cognitive function in different BMI groups. SFAs might increase the risk of cognitive decline, and the intake of n-3 PUFAs could be a protective factor for cognitive function in obese people, while the intake of n-6 PUFAs played a role in leading to poorer memory in the obese people [23]. Our study provided available evidence for the prevention of obesity and cognitive decline through the intake of dietary fatty acids plausibly.

## 2. Materials and Methods

### 2.1. Participants and Group

We conducted a cross-sectional study in three townships (Caiyu town, Weishanzhuang town and Lixian town) of Daxing District, Beijing from November to December in 2020. A total of 1295 participants volunteered to participate in this investigation. After excluding 13 wasting participants and those who (1) had severe thyroid and kidney dysfunction; (2) had a history of cancer disease, encephalitis, head trauma, and other central nervous diseases, or any type of definite mental illness; (3) had reading, hearing, or vision impairments; or were losing self-caring ability, or had miss or doubtful data that must be presented, 1074 participants (aged from 45 to 75) were recruited into the study (Figure 1).

The participants were classified as normal weight (NW; 18.50 ≤ BMI < 24.00), overweight (OW; 24.00 ≤ BMI < 28.00), and obese (OB; 28.00≤ BMI), according to the Chinese Obesity Working Group [24]. Diabetes mellitus was diagnosed by fasting plasma glucose ≥ 7.0 mmol/L, according to the “Guidelines for the Prevention and Treatment of Type 2 Diabetes in China” published by the Diabetes Society of Chinese Medical Association in 2017 [25]. Hypertriglyceridemia was defined as levels of triglyceride in serum ≥ 1.7 mmol/L [26].

The study protocol was approved by the Ethics Review Board of Capital Medical University (Z2019SY011). Written informed consent was obtained from all of the participants.

### 2.2. Physical Examinations and Blood Sample Collection

The height (m), weight (kg), waist circumference (cm), and hip circumference (cm) were measured by professional medical staff, and the participants were required to take off their shoes and coats. Body mass indexes (BMI = weight (kg)/height (m)^2^) and waist–hip ratios (WHR = waist circumference (cm) divided by hip circumference (cm)) were calculated.

The participants were instructed to come with an empty stomach in order to draw fasting peripheral venous blood samples (10 mL), which were numbered according to the order of the patients. The blood samples were collected in 2 kinds of tubes: without anticoagulant to obtain serum (6 mL), and with ethylenediaminetetraacetic acid (EDTA) as the anticoagulant for hemogram in order to collect red blood cells (4 mL).

### 2.3. Questionnaire Survey

In our study, there were three parts to the questionnaire survey, including sociodemographic characteristics, the assessment of cognitive function, and a dietary intake survey.

#### 2.3.1. Sociodemographic Characteristics

Basic information including the name, age, gender, smoking, drinking, exercise, and employment status of the respondents was gathered by direct questioning. The past medical history and family history of probable AD or dementia were also recorded in this part.

#### 2.3.2. Assessment of Cognitive Function

The Chinese versions of the Mini-Mental State Examination (MMSE) and the MoCA scale were used to provide a general cognitive status and evaluate the cognitive function; they were completed by professionally trained investigators [9,27].

The MMSE scale includes orientation (maximum 10 points), memory (maximum 3 points), attention (maximum 5 points), delayed recall (maximum 3 points), language skills (maximum 9 points); the total score of the MMSE is 30. The MOCA scale includes visuospatial function (maximum 5 points), naming (maximum 3 points), attention ability (maximum 6 points), language skills (maximum 3 points), abstract thinking ability (maximum 2 points), memory ability (maximum 5 points), and orientation ability (maximum 6 points), with a total score of 30 points.

#### 2.3.3. Dietary Intake Survey

The dietary intake data were collected using a 34-item food frequency questionnaire (FFQ) by conducting face-to-face interviews to ascertain the average intake frequency in the past year. The FFQ was adapted from the questionnaire used for the Dietary Investigation of Chinese Residents [28]. The frequency of eating (daily, weekly, monthly, and annual) of each food in the FFQ was recorded in an interview by trained investigators. The Chinese food composition tables (the sixth edition) were used to estimate the participants’ nutrient intakes and cumulative average daily intakes. Besides this, the percentage of each fatty acid in the total fatty acids was used to present the dietary intake of fatty acids.

### 2.4. Blood Biochemistry Parameter Detection

Blood samples (6 mL) were collected using the antecubital vein of each participant, and were centrifuged at 3500 r/min for 10 min to obtain the serum. The enzymatic colorimetric method was used to measure the total cholesterol (TC), total triglycerides (TG), high-density lipoprotein cholesterol (HDL-C), and low-density lipoprotein cholesterol (LDL-C) (Cobas C501, Randox, UK). The level of fasting plasma glucose was measured by the hexokinase method (Cobas C501, Randox, UK). By using the immunonephelometry method, the apolipoprotein E (Apo E) was determined (Cobas C501, Randox, UK) [17].

### 2.5. Fatty Acid Analysis

#### 2.5.1. Collection of the Erythrocyte Membranes

The blood samples (4 mL) were collected using tubes with EDTA, and were centrifuged at 3000 r/min for 20 min to precipitate the red blood cells. The plasma was separated, and the fluffy sediment layer on the surface of the precipitated red blood cells was sucked up using pipettes. After the addition of physiological saline and centrifuging at 4000 r/min for 20 min to wash 3 times, the red blood cells were separated. Then, the pre-chilled 10 mmol/L pH7.4 Tris-HCl buffer solution was added at a ratio of 1:40 in the mixture, and was then placed in the refrigerator at 4 °C for 2 h to complete the hemolysis. Then, the mixture was centrifuged at 8000 r/min at 4°C for 20 min to precipitate the red blood cell membrane. The supernatant solution was discarded, and physiological saline was added to the tube to wash the precipitate 3 times; finally, the erythrocyte membrane samples were obtained.

#### 2.5.2. Fatty Acid Analysis

Gas chromatography analysis was performed in order to determine the fatty acid compositions in the erythrocyte membrane. The detailed steps were as follows: 50 mg of the sample was weighed, and mixed with 2 mL boron trifluoride-methanol (SHBJ5851, Sigma-Aldrich, Darmstadt, Germany). Then, the mixture was water bathed for 60 min at 60 °C (Zhongxingweiye, DZKW-4, Beijing, China) after being vortexed and sealed. We waited for the test tube to cool, and then 1 mL n-hexane (171211, Fisher Chemical, Waltham, MA, USA) was added, and the mixture was shaken (MMV-1000 W, EYELA, Tokyo, Japan). Distilled water was added, and the mixture was centrifuged at 4000 r/min for 5 min to extract the upper organic phase for testing. Then, the mixture was added to hexane (1.5 mL) and shaken with a vortex for 1 min immediately [29].

The gas chromatograph (GC-2010, Shimadzu, Kyoto, Japan) was equipped with a 100.0 mm × 0.25 mm (film thickness 0.2 μm) capillary column (TJMF25, Jinteng, Tianjin, China). The chromatographic reference conditions were as follows: the sampler temperature was 270 °C, the detector temperature was 280 °C, the carrier gas was nitrogen, and there was a split ratio of 100:1 and a volume injection of 1.0 µL. The program heating lasted 49 min, with the oven temperature heating from 100 °C to 230 °C. The fatty acids were expressed as the percentage of the fatty acid curve area from the total fatty acid curve areas in the chromatogram. The fatty acid profile in the erythrocyte membrane was reported as the percentage of each fatty acid detected, total SFAs, MUFAs, total n-6 PUFAs, total n-3 PUFAs, n-6 to n-3 fatty acid ratio, and total TFAs. The fatty acid profile of the extractives in the erythrocyte membrane given by gas chromatography is shown in Figure 2.

### 2.6. Statistical Analysis

Epidata3.1 software (The Epi Data Association, Odense, Denmark) was used for the data entry. All of the data received were double-checked, and a consistency test was performed to ensure accuracy. SPSS 26.0 was used to perform the statistical analysis in our study. The continuous variables (non-normal distribution) were presented as the mean values ± standard deviation (M ± SD), and were tested using the Kruskal–Wallis test. The pairwise analyses for the continuous variables were performed using Bonferroni’s multiple comparison test. Chi-squared tests were used for the discontinuous variables, which were represented by percentages (n (%)). Multiple linear regression was used to obtain the coefficients between the fatty acids and cognition scores. Potential confounders including age, gender, BMI, waist–hip ratio, education, culture, smoking, drinking, exercise, history of hypertension, diabetes mellitus, hypertriglyceridemia, and energy intake were also included in the model. *p* < 0.05 was considered to indicate a statistically significant difference.

## 3. Results

### 3.1. Basic Information

In the present study, 1295 adults were invited to participate in the survey; after excluding 13 wasting participants, 49 from the NW group, 100 from the OW group, and 59 from the OB group according to exclusion criteria, a final total of 1074 participants were recruited into the study (Figure 1). The mean age of the OB group was lower than that of the NW and OW groups (*p* < 0.01). Though smoking status was not different between the three groups, the percentage of those who never smoked in the OB group tended to be higher than that in the NW and OW groups. The OB group had a higher percentage in the history of hypertension and hypertriglyceridemia than the NW group (*p* < 0.01) (Table 1).

### 3.2. Blood Biochemistry Parameters for Comparison among the Three Groups

We further explored whether there was any difference in the levels of blood biochemistry parameters in the serum among the three groups, and found that there was no difference in TC and LDL-C among the three groups (Table 2). However, the levels of TG and FBG in the serum in the OB group were significantly higher than those in the NW and OW groups (*p* < 0.01), while the level of Apo E in the serum in the OB group was significantly higher than that in the NW group (*p* < 0.01). Additionally, the levels of HDL-C in the OW and OB groups were significantly lower than those in the NW group (*p* < 0.01).

### 3.3. Cognitive Scores for Comparison among the Three Groups

Next, we used the MMSE and MoCA scales to assess cognitive function. Our study showed the discrimination of the three groups for the scores of total MMSE, MMSE attention, total MoCA, MoCA attention, and MoCA memory (*p* < 0.05). Moreover, compared with the other two groups, the OW tended to have lower cognitive scores (Table 3), especially attention ability (assessed by MMSE) and memory ability (assessed by MoCA). Apart from these, although there was no significant difference between the NW and OB groups, there was a trend towards lower values for the total MMSE and MoCA scores in the OB group compared to the scores in the NW group.

### 3.4. The Fatty Acid Composition of the Erythrocyte Membranes for Comparison among the Three Groups

As shown in Appendix A, there were no significant differences of the intake of SFAs, MUFAs and PUFAs among the three groups, while the intake of fatty acids that was unknown (UN-Ks) in the OB group was higher than that in the OW group. However, some fatty acids were not detected exactly, and we could not distinguish n-3 PUFA and n-6 PUFAs using the Chinese food composition tables (the sixth edition). In order to evaluate the differences of fatty acid intake, gas chromatography analysis was performed to test the fatty acid composition of erythrocyte membranes; this found that the proportions of SFAs in the erythrocyte membrane fatty acid profiles in the OB group were higher than those in the OW and NW groups, such as C16:0 and total SFAs, except for C17:0. The level of C18:0 in the erythrocyte membranes in the obese participants was higher than that in the overweight participants. Opposed to the SFAs was the fact that the proportions of MUFAs and PUFAs in the erythrocyte membrane fatty acid profile in OB were lower than in the NW and OW groups, especially C17:1, C18:1, C18:2 n-6, C20:4 n-6, total n-6 PUFAs, and total PUFAs (*p* < 0.05). However, the proportion of total n-3 PUFAs in the OB group was higher than in OW group (Table 4).

### 3.5. Associations between the Scores in All of the Cognitive Domains Measured by MMSE or MoCA and the Fatty Acid Composition of the Erythrocyte Membranes in Different Groups

In order to further define the influence of the intake of fatty acids on cognitive function, statistical analyses were performed via multiple linear regression. By analyzing the results shown in Appendix A, we found—in all three groups—that dietary SFAs exhibited a significant negative correlation with cognitive scores. MUFAs were significantly positively correlated with cognitive functions (memory ability and language skills) in the NW group. The associations of PUFAs with the scores of different cognitive functions were inconsistent. In particular, higher PUFAs could be a predictor for the decline of the abstracting function in all three groups, while the increase of PUFAs in the diet accompanied the improvement of the score of visuospatial function in those who were of normal weight and obese. The relationship between the fatty acid composition of the erythrocyte membranes and the cognitive function scores was also assessed to obtain a more accurate result regarding the impact of fatty acid intake on cognitive function. From the result displayed in Figure 3, and Table 5 and Table 6, in all participants, regardless of BMI groups, almost all of the SFAs were significantly negatively correlated with the cognitive function scores, especially C10:0, C11:0, C16:0, C17:0, and C24:0, except for C18:0 in the NW group and C13:0 in the OB group (*p* < 0.05). C20:0 in the OB group was negatively related to MoCA naming (*p* < 0.05), while it was positively related to MoCA language skills (*p* < 0.05), as well as to the relationship between C20:0 and MMSE attention in the NW group. The MUFA distribution (C18:1 n-9c in the NW group, C16:1 and C17:1 in the OW group, and C22:1n-9 in the OB group) in the erythrocyte membranes was negatively related to cognitive functions, as assessed by MMSE and MoCA (*p* < 0.05), while a higher proportion of C20:1 played a positive role in the increase of cognitive functions assessed by MoCA (*p* < 0.05), such as the association between C20:1 and language skill assessed by MoCA in the OW group, and the relationship between C20:1 and the scores of total MoCA and MoCA visuospatial function. Besides this, the level of total PUFAs was significantly negatively correlated with the cognitive function scores in both the NW and OW groups (*p* < 0.01), and C18:2 n-6 also had negative relationship with language skill, as tested by MoCA in the NW group (*p* < 0.01), as well as C18:2 n-6 with the MMSE delayed recall score in the OB group. Total n-6 PUFAs were negatively correlated with the scores of MoCA abstracting and MoCA orientation in the NW group (*p* < 0.05). Furthermore, C20:4 n-6 was negatively related to the scores of the cognitive functions in all three groups. Higher levels of C20:3n-3, C20:5n-3 and total n-3 PUFAs were accompanied by worse cognitive function in the OW group, while n-6/n-3 PUFAs were positively related to the MMSE delayed recall in the OW group (*p* < 0.01). In contrast, higher levels of C18:3 n-3, C20:5 n-3 and total n-3 PUFAs in the erythrocyte membranes were accompanied by better cognitive function in the OB group, especially attention function, as tested by MoCA (*p* < 0.05).

## 4. Discussion

In this study, a cross-sectional study was conducted to assess the relationship between dietary fatty acid intake, the fatty acid profile in the erythrocyte membrane, and cognitive function—as evaluated by MMSE and MoCA scales—in the overweight and obese populations.

The higher percentages in the history of hypertension and hypertriglyceridemia, and the higher TG, FBG, and Apo E levels in the serum in the OB group confirmed that obesity could be a risk factor for chronic diseases [2,30,31], which is consistent with previous studies. Given that the space and time of the survey were limited, instead of measuring blood pressure at the measurement site, the history of hypertension was asked face-to-face. There might be some recall bias that influenced results.

Several studies have reported that the increase in BMI was closely associated with cognitive damage [32,33]. Our results contrasted slightly with previous research. Our studies suggested that the cognitive function scores in the OW groups were the lowest among the three groups. Though there were no significant differences between the NW and the OB groups, the cognitive functions of the obese participants tended to be worse compared with the normal weight participants. Ribeiro concluded that severe obesity might lead to cognitive decline [2]. However, in a study conducted in villages South Africa, higher cognitive function scores were accompanied with higher BMI [34]. Other surveys conducted in Asia also exhibited different interpretations. A study carried out in the elderly Chinese population in Chongqing pointed out that being overweight was significantly associated with a decreased risk of cognitive impairment, and abdominal obesity was a risk factor of cognitive decline after adjusting the covariates [35]. Results from Korea considered that a high BMI could be a protective factor for cognitive dysfunction in older adulthood, although as participants aged, the cognitive function become substantially worse [36]. This might result from the education levels of the participants, as most of the subjects in this study had lower education ages than junior high school, while a previous study suggested that lower education ages could lead to poorer cognitive performance [17]. In addition, participants might report a higher level of education. Previous studies also indicated that some chronic diseases might be risk factors for cognitive decline, such as diabetes mellitus, hypertriglyceridemia and hypertension, which might be the reasons that the OW and OB group tend to obtain lower scores for their cognitive functions [37,38], and that higher proportions of hypertriglyceridemia and hypertension were shown in OB and OW groups. Furthermore, the bodyweight in the whole life course might contribute to different changes in overall cognitive performance, which was proposed by Zhou, who suggested that a high BMI in early adult life was associated with low cognitive function in late life [39]. Another reason might be that obesity might be a crucial factor [40], as a greater waist–hip ratio was directly associated with worse global cognitive performance. Furthermore, the results obtained from the past studies showed that those who were currently smoking or had ever smoked had poorer cognitive performance [41], and age was independently associated with faster cognitive decline [3]. On the other hand, the participants in the OB group in our study had a lower average age and percentage of current smoking or ever having smoked. This might be an important cause of the fact that no significant difference was identified in the cognitive function scores between the OB and the NW groups.

The traditional dietary intake survey, collected by using a 34-item FFQ through face-to-face interviews, held several limitations, such as recall bias and food category limitations. Furthermore, the Chinese food composition tables could not provide whole dietary fatty acids, such as TFAs, and could not distinguish n-3 PUFAs and n-6 PUFAs. Based on these rationales, gas chromatography analysis was performed to determine the fatty acid compositions in the erythrocyte membrane, which are relatively stable and could affect long-term fatty acid intake, such that could be used to reflect the dietary fatty acid intake levels in the past few months [21,42], could provide information on the lipid balance and metabolism in the body, and could act as biomarkers for various metabolic diseases including obesity, diabetes, metabolic syndrome, etc. [43,44].

A previous study indicated that long-term and high SFA intake could increase AD morbidity and exacerbate symptoms of dementia [45]. Researchers [46] found that tea seed oil that was rich in MUFAs could prevent obesity, compared with either tea seed oil rich in PUFA or tea seed oil rich in SFA, in HFD-induced obese mice. However, another study also showed that high-SFA and high-PUFA overfeeding both resulted in a similar body weight increase [47], which meant that weight could be affected not solely by fatty acids types; therefore, the intake of fatty acids should also be taken into consideration. Previous studies have confirmed that a higher SFA intake is related to worse global cognitive and memory [48], and a higher SFA percentage in erythrocyte membranes fatty acids was found in AD patients than in the healthy individuals of the control group [49]. In all of the groups, almost all of the SFAs intake was negatively associated with cognitive function scores, no matter the dietary fatty acids or erythrocyte membrane fatty acids in our study. In particular, higher C10:0, C11:0, C16:0, C17:0 and C24:0 tended to associated with worse cognitive function, which was consistent with the previous results [23]; our study pointed out that higher SFAs proportions in the erythrocyte membrane fatty acid composition could perhaps predict cognitive impairment in the middle-aged and elderly in rural China. Conversely, previous studies showed that the association closest to significance was negative, i.e., that between erythrocyte membrane stearic acid (C18:0) and amyloid-β (Aβ) [50] and a greater risk of cognitive impairment [51]. Opposed to these results, no matter the higher level of C18:0 in the NW group or the higher levels of C13:0 and C20:0 in the erythrocyte membranes, the OB group had a preference for better cognitive function in our study. As a medium-chain fatty acid (MCFA) [52], lauric acid (C12:0) could prevent obesity [53] and reduce neuroinflammatory responses [54], which might be the reasons associated with the higher cognitive function [55]. Accordingly, it could be speculated that tridecyl acid (C13:0) might also act as a protective factor for cognitive function. This might be affected by body fat. However, in our study, C18:0 intake exhibited a positive correlation with MoCA abstracting in the NW group. There were also studies showing the results consistent with our study [51]. This disparity was likely because of its source and metabolism in the body. A recent study showed that C18:0 from *Alyssum homolocarpum* seed oil (AHSO) [56] and palmitic acid hydroxy stearic acid (5-PAHSA) [57] could prevent cognitive decline. Based on this, an in-depth investigation of their dietary sources was needed. Combined with the result that the C16:0, C17:0 and the total SFAs percentages of the fatty acid composition of erythrocyte membranes in the OB were higher than those in the OW and NW groups, while the scores of total MMSE and MoCA in the OB tended to be lower than in the NW group, we could draw a conclusion that a high-SFA diet might lead to the increase in weight and cognitive decline in obesity.

Although a higher intake of total MUFAs was positively related to cognitive functions, our study showed that higher levels of C16:1, C17:1, and C22:1n-9 in the erythrocyte membranes could result in worse memory ability, naming ability, attention ability and language skills in the OW and OB group, while C20:1 played a contrast role in this process with lower percentages of MUFAs in the erythrocyte membrane fatty acid profile in the OB group than in the other two groups. This was not consistent overall with previous findings. Several studies have indicated that, instead, a diet rich in MUFAs resulted in a reduced risk of neurodegenerative diseases, such as AD [58], and greater global cognitive function [48]. On the other hand, studies also revealed a negative association between the level of erucic acid (C22:1n-9) in the erythrocyte membrane and the number-combination test result [59]. In addition, researchers found that the level of C16:1 in the peripheral serum in normal individuals was lower than those in the MCI and AD participants [60]. We could conclude that the intake of MUFAs might be a protective factor for obesity prevention in the middle-aged and older adults, and could reduce the impairment of cognitive function for the normal weight participants, while C20:1 could have beneficial effects on the cognitive function improvement in participants who were overweight or obese. We could only speculate regarding the reason as to why n-9 fatty acids were negatively correlated with leptin gene expression in visceral adipose [61] in obesity, or regarding the interaction between different fatty acids.

As for the PUFAs, C18:2 n-6, C20:3 n-6, C20:4 n-6, total n-6 PUFAs, C22:6 n-3, and the total levels of PUFAs in the erythrocyte membranes in the OB group were lower than those in the other groups; however, the level of total n-3 PUFAs in the OB group was higher than that in the OW group, and the α-linolenic acid (C18:3n-3; ALA) and C20:3 n-3 in the erythrocyte membranes also tended to be higher in the OB group than in the OW group. Besides this, n-6 PUFAs showed negative associations with cognitive functions (languages skills, delayed recall ability, and memory ability) in all of the participants, while a higher n-6/n-3 PUFAs ratio resulted in better delayed recall ability in the OW group. Our findings were not fully consistent with those in previous studies, in which lower PUFAs in the erythrocyte membrane were accounted for as a predictor of cognitive decline [23]. However, in our study, the level of total PUFAs in the erythrocyte membrane was negatively related to scores of cognitive functions (such as memory and orientation) in the NW and OW group. The intake ratio of A to B might explain this phenomenon, because higher n-6/n-3 PUFAs was linked to better delayed recall ability, as assessed by MMSE, in the OW group. In addition, n-3 PUFAs played diametrically roles in the OW and the OB groups. In the OB group, n-3 PUFAs might be a protective factor for cognitive function, though they had a negative impact in the OW group. Recent studies showed that n-3 PUFAs might have preventive effects on cognitive function [62]. A Boston Puerto Rican Health Study (aged 57 years) also indicated that the total n-3 very-long-chain fatty acids (VLCFAs) in erythrocyte and the diet were associated with better executive function, while lower MMSE and executive function scores were associated with greatern-6 PUFAs [63]. There was a study which also suggested that the supplementation of n-3 PUFAs could improve the memory ability of male Long-Evans rats by inhibiting the neuroinflammation which was induced by interleukin-1β (IL-1β) [64], and that the supplementation of ALA could increase cognitive function [65]. In addition, compared with patients with mild neurocognitive impairment, the EPA and DHA (transformed from ALA [66]) levels in the plasma of patients with severe neurocognitive impairment were lower [67]. Another study also revealed that ALA and DHA protected microglia against a decrease in cell viability in obese people with cognitive impairment [68]. The above results were partially in line with our study, as a study in older Australian adults suggested that a high intake of n-6 PUFAs could improve cognitive function [69]. Heude et al. also found that the percentage of n-6 PUFAs increasing and the decrease of percentage of n-3 PUFAs were associated with a greater risk of cognitive decline [51]. However, a study of older participants found that EPA and DHA were negatively linked to MMSE score [70]. However, as most of the previous studies did not distinguish the overweight and obese participants, or only normal weight subjects were included in the study, we could not obtain more evidence of the effects of the intake of fatty acids on cognitive function in different BMI individuals. However, we confirmed that n-3 PUFAs, especially ALA and EPA, could protect cognitive function in obesity. In addition, the proportions of the overweight and the obese in out of all of the participants might be an important reason that led to this difference, as some studies did not classify the participants according to BMI, or the proportions of the overweight and the obese were small in these studies. Though it was also shown that high TFA intake was implicated in the development and progression of obesity [44,45], there was no significant difference in TFA intake and the percentage of TFAs in erythrocyte membrane fatty acids among the three groups. We conjecture that this was due to less-frequently consumed cream cakes, cookies, or fried foods that were rich in TFAs in the subjects of the present study.

Some limitations existed in our estimate. The dietary intake of fatty acids was affected by novel Coronavirus pneumonia in 2020, Beijing; a small portion of participants changed their dietary habits consciously with less seafood consumed than before, which could provide rich PUFAs, although that was a very small decrease. Besides this, our study sample was relatively small. Moreover, the samples in the three groups were different, which might result in a certain deviation. As our participants came from urban areas of Beijing, our results were not necessarily representative of the whole of Beijing. More studies are needed to find the exact effect of dietary fatty acids on cognitive function, and the possible mechanisms of cognitive change due to the intake of fatty acids in the overweight and obese population.

## 5. Conclusions

High SFA intake could play a key role in cognitive decline in all individuals, MUFAs might prevent those who were of normal weight from cognitive decline, and n-3 PUFA intake had a protective role in cognitive decline in those who were obese, while PUFAs might increase the risk of cognitive impairment in the overweight population. Dietary fatty acid intake had different effects on the cognitive function of overweight and obese people; more precise dietary advice is needed to prevent cognitive impairment. Available data were plausibly provided for the prevention of obesity and cognitive decline by the rational intake of dietary fatty acids. The findings of our study provided relevant practical implications to the field for both researchers and practitioners, such as the provision of more accurate and individual food-based dietary guidance for overweight and obese people.

## Figures and Tables

**Figure 1 nutrients-14-00914-f001:**
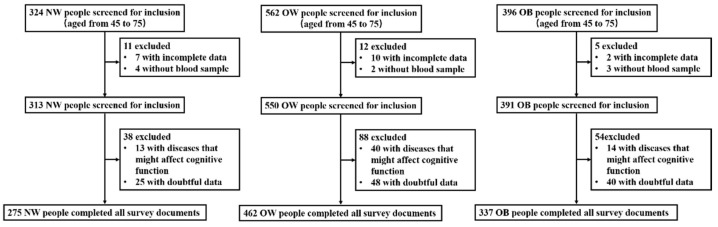
Study flowchart. NW: normal weight group; OW: overweight group; OB: obese group.

**Figure 2 nutrients-14-00914-f002:**
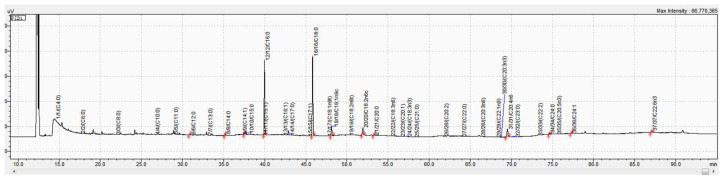
Human erythrocyte membrane fatty acid standard spectrum.

**Figure 3 nutrients-14-00914-f003:**
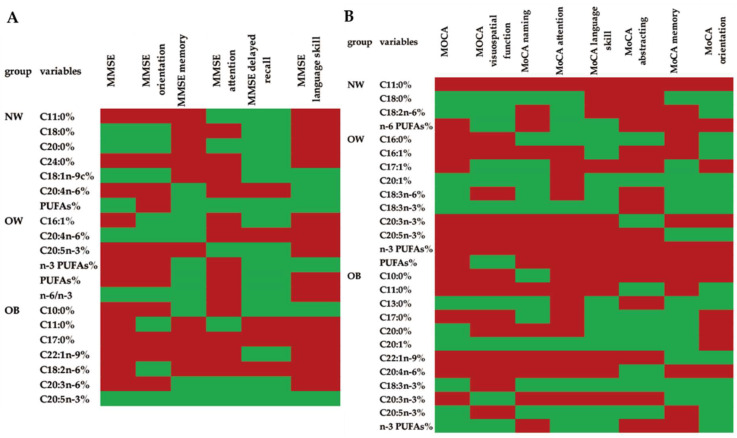
Correlations between the fatty acid compositions of the erythrocyte membranes in the three groups and their cognitive functions scores: heatmaps of the regression coefficients in a multiple linear regression. (**A**) Correlations between the fatty acid composition of the erythrocyte membranes in the three groups and the cognitive functions scores determined by MMSE. (**B**) Correlations between the fatty acid composition of the erythrocyte membranes in the three groups and the cognitive functions scores determined by MoCA. The red color indicates a negative correlation, while the green color indicates a positive correlation.

**Table 1 nutrients-14-00914-t001:** Demographics of the study population.

		NW (n = 275)	OW (n = 462)	OB (n = 337)	*p*
Continuous variable		M ± SD	M ± SD	M ± SD	
Age		61.476 ± 7.493	60.738 ± 6.937	59.362 ± 6.885 ^ab^	0.001 **
BMI (kg/m^2^)		22.142 ± 1.365	25.935 ± 1.131 ^a^	30.819 ± 2.430 ^ab^	<0.001 **
Waist-hip ratio		0.898 ± 0.066	0.910 ± 0.057	0.923 ± 0.066 ^a^	0.001 **
Energy intake (kcal)		1789.213 ± 1041.815	1747.080 ± 940.751	1884.528 ± 922.536	0.126
Categorical variable		n (%)	n (%)	n (%)	
Gender	male	94 (34.18%)	158 (34.20%)	103 (30.56%)	0.503
	female	181 (65.82%)	304 (65.80%)	234 (69.44%)	
Culture	illiterate	16 (5.86%)	30 (6.52%)	27 (8.06%)	0.961
	primary school	64 (23.44%)	108 (23.48%)	71 (21.19%)	
	junior high school	142 (52.01%)	239 (51.96%)	175 (52.24%)	
	senior middle school	45 (16.48%)	72 (15.65%)	50 (14.93%)	
	technical secondary school	3 (1.10%)	5 (1.09%)	6 (1.79%)	
	junior college	1 (0.37%)	3 (0.65%)	5 (1.49%)	
	undergraduate or above	2 (0.73%)	3 (0.65%)	1 (0.30%)	
Smoking	never	194 (72.66%)	346 (76.38%)	269 (80.54%)	0.208
	continuous smoking for at least 6 months	67 (25.09%)	97 (21.41%)	57 (17.07%)	
	smoking for at least 6 months but not continuous	6 (2.25%)	10 (2.21%)	8 (2.40%)	
Drinking	never	197 (73.23%)	334 (73.25%)	235 (71.00%)	0.836
	former	10 (3.72%)	15 (3.29%)	16 (4.83%)	
	current	62 (23.05%)	107 (23.46%)	80 (24.17%)	
Exercise	no	36 (13.14%)	57 (12.47%)	44 (13.29%)	0.935
	yes	238 (86.86%)	400 (87.53%)	287 (86.71%)	
History of hypertension	no	180 (65.45%)	238 (51.52%) ^a^	154 (45.70%) ^a^	<0.001 **
	yes	91 (33.09%)	223 (48.27%) ^a^	181 (53.71%) ^a^	
	unknow	4 (1.45%)	1 (0.22%) ^a^	2 (0.59%)	
Diabetes mellitus	no	228 (82.91%)	390 (84.42%)	264 (78.34%)	0.080
	yes	47 (17.09%)	72 (15.58%)	73 (21.66%)	
Hypertriglyceridemia	no	210 (76.36%)	290 (62.77%) ^a^	192 (56.97%) ^a^	<0.001 **
	yes	65 (23.64)	172 (37.23%) ^a^	145 (43.03%) ^a^	

Data are M ± SD or n (%). The continuous variable was tested with the Kruskal–Wallis test. The discontinuous variable was tested with the Chi-square test. NW: normal weight group; OW: overweight group; OB: obese group; BMI: body mass index; ** *p* <0.01. ^a^ indicates a significant difference compared with the normal weight group; ^b^ indicates a significant difference compared with the overweight group.

**Table 2 nutrients-14-00914-t002:** Blood biochemistry parameters of the study population in the three groups.

Variables	NW (n = 275)	OW (n = 462)	OB (n = 337)	*p*
TC (mmol/L)	5.311 ± 1.030	5.210 ± 1.052	5.190 ± 1.066	0.295
TG (mmol/L)	1.379 ± 1.004	1.831 ± 1.654 ^a^	1.895 ± 1.181 ^ab^	<0.001 **
HDL-C (mmol/L)	1.515 ± 0.326	1.391 ± 0.309 ^a^	1.343 ± 0.295 ^a^	<0.001 **
LDL-C (mmol/L)	3.168 ± 0.807	3.166 ± 0.791	3.214 ± 0.838	0.633
FBG (mmol/L)	6.066 ± 2.916	5.932 ± 2.423	6.335 ± 2.388 ^ab^	<0.001 **
Apo E (mg/L)	51.778 ± 14.882	54.906 ± 18.927	55.996 ± 17.280 ^a^	0.019 **

Data are M ± SD. The continuous variable was tested with the Kruskal–Wallis test. NW: normal weight group; OW: overweight group; OB: obese group; TC: total cholesterol; TG: total triglycerides; HDL-C: high density lipoprotein cholesterol; LDL-C: low density lipoprotein cholesterol; FBG: fasting blood glucose; Apo E: apolipoprotein E; ** *p* < 0.01. ^a^ indicates a significant difference compared with the normal weight group; ^b^ indicates a significant difference compared with the overweight group.

**Table 3 nutrients-14-00914-t003:** Cognitive score of the study population in the three groups.

Variables	NW (n = 275)	OW (n = 462)	OB (n = 337)	*p*
MMSE	26.920 ± 2.843	26.409 ± 2.996	26.751 ± 2.826	0.049 *
MMSE orientation	9.447 ± 0.996	9.429 ± 0.916	9.504 ± 0.890	0.224
MMSE memory	2.865 ± 0.419	2.846 ± 0.422	2.905 ± 0.349	0.054
MMSE attention	4.047 ± 1.346	3.745 ± 1.447 ^a^	3.878 ± 1.367	0.014 *
MMSE delayed recall	2.284 ± 0.858	2.232 ± 0.899	2.303 ± 0.858	0.532
MMSE language skills	8.262 ± 1.048	8.180 ± 1.016	8.157 ± 1.016	0.136
MOCA	21.567 ± 4.235	20.781 ± 4.142 ^a^	21.478 ± 4.075	0.011 *
MOCA visuospatial function	2.967 ± 1.265	2.879 ± 1.224	2.997 ± 1.252	0.271
MoCA naming	2.738 ± 0.570	2.721 ± 0.568	2.721 ± 0.587	0.843
MoCA attention	5.102 ± 1.103	4.970 ± 1.140	5.169 ± 0.990	0.044 *
MoCA language skills	1.982 ± 0.949	1.907 ± 0.914	1.958 ± 0.941	0.477
MoCA abstracting	1.007 ± 0.863	0.937 ± 0.860	0.935 ± 0.857	0.498
MoCA memory	2.193 ± 1.671	1.829 ± 1.608 ^a^	2.119 ± 1.582 ^b^	0.005 **
MoCA orientation	5.578 ± 0.786	5.539 ± 0.741	5.579 ± 0.752	0.311

Data are M ± SD. The continuous variable was tested with the Kruskal–Wallis test. NW: normal weight group; OW: overweight group; OB: obese group; MMSE: Mini-Mental State Examination; MoCA: Montreal Cognitive Assessment; * *p* < 0.05; ** *p* < 0.01. ^a^ indicates a significant difference compared with the normal weight group; ^b^ indicates a significant difference compared with the overweight group.

**Table 4 nutrients-14-00914-t004:** The fatty acid composition of the erythrocyte membranes of the study population in the three groups.

Variables	NW (n = 275)	OW (n = 462)	OB (n = 337)	*p*
C4:0%	0.000 ± 0.000	0.000 ± 0.000	0.000 ± 0.000	1.000
C6:0%	0.000 ± 0.000	0.000 ± 0.000	0.000 ± 0.000	1.000
C8:0%	0.000 ± 0.000	0.000 ± 0.000	0.000 ± 0.000	1.000
C10:0%	0.000 ± 0.000	0.000 ± 0.000	0.003 ± 0.059	0.335
C11:0%	0.139 ± 2.299	0.139 ± 2.245	0.235 ± 2.379	0.138
C12:0%	0.238 ± 0.703	0.217 ± 0.659	0.370 ± 0.898	0.130
C13:0%	0.069 ± 0.340	0.079 ± 0.461	0.164 ± 0.715	0.109
C14:0%	0.239 ± 0.452	0.214 ± 0.434	0.334 ± 0.673	0.251
C15:0%	0.859 ± 1.251	1.010 ± 1.921	1.088 ± 2.172	0.289
C16:0%	27.651 ± 5.262	28.636 ± 5.032 ^a^	29.333 ± 5.852 ^ab^	<0.001 **
C17:0%	0.132 ± 0.272	0.168 ± 0.591	0.098 ± 0.273 ^ab^	0.001 **
C18:0%	17.880 ± 4.959	17.667 ± 5.058	18.836 ± 5.363 ^b^	0.015 *
C20:0%	0.017 ± 0.099	0.007 ± 0.070	0.008 ± 0.064	0.176
C21:0%	0.000 ± 0.000	0.000 ± 0.000	0.001 ± 0.020	0.335
C22:0%	0.000 ± 0.000	0.000 ± 0.000	0.000 ± 0.000	1.000
C23:0%	0.010 ± 0.102	0.000 ± 0.000	0.007 ± 0.069	0.071
C24:0%	0.038 ± 0.156	0.042 ± 0.176	0.038 ± 0.191	0.448
SFAs%	47.273 ± 8.336	48.179 ± 9.219	50.516 ± 9.989 ^ab^	0.001 **
C14:1%	0.000 ± 0.000	0.011 ± 0.110	0.017 ± 0.165	0.121
C15:1%	0.018 ± 0.158	0.022 ± 0.182	0.039 ± 0.243	0.414
C16:1%	0.099 ± 0.148	0.077 ± 0.125	0.073 ± 0.143 ^a^	0.024 *
C17:1%	0.016 ± 0.058	0.021 ± 0.109	0.008 ± 0.038 ^b^	0.009 **
C18:1n-9%	11.172 ± 1.775	10.983 ± 2.149	10.547 ± 2.371 ^ab^	<0.001 **
C20:1%	0.021 ± 0.175	0.026 ± 0.117	0.057 ± 0.228	0.375
C22:1n-9%	0.000 ± 0.000	0.001 ± 0.011	0.001 ± 0.020	0.683
C24:1%	0.942 ± 1.797	0.832 ± 1.593	1.089 ± 2.205	0.615
MUFAs%	12.268 ± 2.301	11.971 ± 2.616	11.831 ± 2.961	0.051
C18:2n-6%	13.703 ± 2.746	13.417 ± 3.068	12.788 ± 3.210 ^ab^	0.002 **
C18:3n-6%	0.001 ± 0.011	0.004 ± 0.043	0.010 ± 0.083	0.322
C20:3n-6%	1.283 ± 0.607	1.249 ± 0.677	1.128 ± 0.681 ^b^	0.028 *
C20:4n-6%	20.488 ± 4.356	20.113 ± 4.612	18.579 ± 5.384 ^ab^	<0.001 **
n-6 PUFAs%	35.474 ± 6.398	34.784 ± 7.093	32.504 ± 8.040 ^ab^	<0.001 **
C18:3n-3%	0.032 ± 0.153	0.036 ± 0.190	0.057 ± 0.299	0.918
C20:3n-3%	0.091 ± 1.502	0.107 ± 1.623	0.454 ± 3.162	0.031 *
C20:5n-3%	0.006 ± 0.074	0.003 ± 0.041	0.002 ± 0.030	0.979
C22:6n-3%	4.816 ± 1.379	4.877 ± 1.458	4.605 ± 1.558 ^ab^	0.005 **
n-3 PUFAs%	4.944 ± 2.059	5.023 ± 2.037	5.118 ± 3.743 ^b^	0.015 *
C20:2%	0.041 ± 0.113	0.042 ± 0.116	0.032 ± 0.101	0.385
C22:2%	0.000 ± 0.000	0.000 ± 0.000	0.000 ± 0.000	1.000
PUFAs%	40.459 ± 7.128	39.848 ± 7.938	37.653 ± 8.622 ^ab^	<0.001 **
C18:1n-9 t %	0.000 ± 0.000	0.001 ± 0.026	0.000 ± 0.000	0.512
C18:2n-6t%	0.000 ± 0.000	0.000 ± 0.000	0.000 ± 0.000	1.000
TFAs%	0.000 ± 0.000	0.001 ± 0.026	0.000 ± 0.000	0.512
n-6/n-3	7.708 ± 2.155	7.321 ± 1.755	7.220 ± 2.000	0.061

Data are M ± SD. The continuous variable was tested with the Kruskal–Wallis test. NW: normal weight group; OW: overweight group; OB: obese group; SFAs: saturated fatty acids; MUFAs: monounsaturated fatty acid; PUFAs: polyunsaturated fatty acids; TFAs: trans fatty acids; * *p* < 0.05; ** *p* < 0.01. ^a^ indicates a significant difference compared with the normal weight group; ^b^ indicates a significant difference compared with the overweight group.

**Table 5 nutrients-14-00914-t005:** Associations between the scores in all of the cognitive domains measured by MMSE and the fatty acid composition of the erythrocyte membranes in the three groups.

	Variables	MMSE		MMSE Orientation		MMSE Memory		MMSE Attention		MMSE Delayed Recall		MMSE Language Skills	
		B	*p*	B	*p*	B	*p*	B	*p*	B	*p*	B	*p*
NW													
	C11:0%	−0.011	0.846	−0.094	0.114	−0.026	0.015 *	0.030	0.589	0.050	0.393	−0.044	0.443
	C18:0%	0.064	0.048 *	0.043	0.656	−0.041	0.506	−0.018	0.758	0.004	0.957	−0.018	0.765
	C20:0%	0.089	0.125	0.051	0.412	−0.098	0.103	1.618	0.037 *	0.028	0.650	−0.016	0.781
	C24:0%	−0.078	0.166	−0.070	0.237	−0.033	0.570	−1.117	0.021 *	0.058	0.319	−0.039	0.496
	C18:1n-9c%	0.042	0.458	0.062	0.344	−0.055	<0.001 **	−0.009	0.873	0.049	0.416	0.076	0.188
	C20:4n-6%	−0.040	0.600	−0.044	0.695	0.022	0.723	−0.008	0.892	−0.031	0.007 **	0.091	0.117
	PUFAs%	0.013	0.882	−0.021	0.011 *	0.029	0.661	0.032	0.599	0.093	0.402	0.073	0.207
OW													
	C16:1%	−0.050	0.287	0.031	0.541	0.020	0.674	−1.198	0.028 *	0.026	0.594	−0.028	0.527
	C20:4n-6%	0.044	0.599	0.049	0.584	0.018	0.703	−0.029	0.540	−0.030	0.001 **	−0.037	0.403
	C20:5n-3%	−0.024	0.586	−0.012	0.804	−0.009	0.839	0.032	0.486	0.007	0.884	−2.235	0.033 *
	n-3 PUFAs%	−0.061	0.218	−0.069	0.194	0.040	0.390	−0.081	0.017 *	0.050	0.411	0.002	0.956
	PUFAs%	−0.061	<0.001 **	−0.021	<0.001 **	0.050	0.285	−0.059	0.285	0.056	0.538	−0.036	0.403
	n-6/n-3	0.068	0.119	0.035	0.443	0.003	0.942	−0.035	0.547	0.065	0.007 **	−0.005	0.908
OB													
	C10:0%	−0.066	0.182	−0.087	0.097	0.011	0.827	−3.169	0.010 *	0.045	0.402	0.039	0.434
	C11:0%	−0.064	0.190	0.037	0.486	−0.031	<0.001 **	0.019	0.724	−0.061	0.001 **	−0.023	0.646
	C17:0%	−0.070	0.155	−0.434	0.011 *	−0.017	0.744	−0.012	0.821	−0.037	0.500	−0.034	0.503
	C22:1n-9%	−0.036	0.471	−0.025	0.631	−2.658	0.004 **	−0.028	0.603	0.037	0.495	−0.041	0.413
	C18:2n-6%	−0.084	0.088	0.049	0.421	−0.021	0.682	−0.064	0.238	−0.033	0.024 *	−0.046	0.359
	C20:3n-6%	−0.042	0.400	−0.142	0.048 *	0.031	0.545	0.039	0.475	0.020	0.748	−0.002	0.966
	C20:5n-3%	9.965	0.030 *	0.074	0.163	0.071	0.171	0.074	0.173	0.098	0.073	0.023	0.643

Multiple linear regression adjusted for age, gender, BMI, waist–hip ratio, culture, smoking, drinking, exercise, history of hypertension, diabetes mellitus, hypertriglyceridemia, and energy intake. NW: normal weight group; OW: overweight group; OB: obese group; PUFAs: polyunsaturated fatty acids; MMSE: Mini-Mental State Examination; * *p* < 0.05; ** *p* < 0.01.

**Table 6 nutrients-14-00914-t006:** Associations between the scores in all of the cognitive domains measured by MoCA and the fatty acid composition of the erythrocyte membranes in the three groups.

	Variables	MoCA		MoCA Visuospatial Function		MoCA Naming		MoCA Attention		MoCA Language Skills		MoCA Abstracting		MoCA Memory		MoCA Orientation	
		B	*p*	B	*p*	B	*p*	B	*p*	B	*p*	B	*p*	B	*p*	B	*p*
NW																	
	C11:0%	−0.095	0.065	−0.028	0.621	−0.040	0.006 **	−0.039	0.490	−0.052	0.022 *	−0.006	0.913	−0.058	0.286	−0.050	0.014 *
	C18:0%	0.109	0.015 *	0.033	0.567	0.035	0.562	0.053	0.354	−0.089	0.200	−0.086	0.335	0.088	0.108	0.028	0.767
	C18:2n-6%	0.050	0.425	0.108	0.060	−0.035	0.548	0.094	0.095	−0.068	<0.001 **	−0.030	0.766	−0.060	0.276	0.068	0.499
	n-6 PUFAs%	−0.003	0.971	0.043	0.459	−0.066	0.268	0.003	0.965	0.101	0.278	−0.019	0.020 *	−0.066	0.228	−0.023	0.002 **
OW																	
	C16:0%	−0.058	0.385	−0.016	0.725	0.062	0.184	0.024	0.605	0.049	0.323	0.055	0.230	−0.055	0.018*	0.042	0.576
	C16:1%	−0.072	0.112	−1.273	0.004 **	−0.068	0.140	−0.930	0.023 *	0.041	0.388	−0.012	0.789	−0.045	0.351	0.045	0.370
	C17:1%	−0.050	0.230	0.033	0.459	0.025	0.595	−1.424	0.002 **	−0.049	0.279	−0.035	0.457	0.002	0.955	−0.020	0.672
	C20:1%	0.073	0.077	0.041	0.350	0.063	0.171	−0.043	0.326	0.771	0.027 *	0.020	0.664	0.073	0.106	0.022	0.639
	C18:3n-6%	0.005	0.901	−0.067	0.138	0.035	0.456	−0.020	0.648	2.071	0.030 *	−0.012	0.791	0.008	0.849	0.026	0.576
	C18:3n-3%	0.066	0.112	0.036	0.413	0.041	0.379	0.013	0.777	0.064	0.168	−0.037	0.423	0.735	0.046 *	0.015	0.748
	C20:3n-3%	−0.260	0.012 *	−0.015	0.738	−0.031	0.504	−0.080	0.010 *	−0.004	0.945	0.002	0.961	−0.045	0.308	−0.051	0.270
	C20:5n-3%	−0.061	0.139	−0.066	0.135	−0.046	0.326	−0.013	0.766	−2.636	0.008 **	−0.008	0.863	0.029	0.515	0.022	0.633
	n-3 PUFAs%	−0.031	0.700	−0.035	0.449	−0.043	0.355	−0.010	0.881	−0.066	0.001 **	−0.038	0.408	−0.031	0.536	−0.052	0.327
	PUFAs%	−0.091	<0.001**	0.029	0.547	−0.088	0.056	−0.030	0.535	−0.032	0.542	−0.038	0.415	−0.060	<0.001 **	−0.019	<0.001 **
OB																	
	C10:0%	−0.094	0.061	−0.038	0.482	0.018	0.742	−2.022	0.018 *	−0.061	0.267	−0.012	0.825	−0.030	0.583	−0.104	0.053
	C11:0%	−0.072	0.141	−0.038	0.472	−0.074	0.174	−0.033	0.530	−0.065	0.232	0.027	0.615	−0.070	0.045 *	0.047	0.372
	C13:0%	0.065	0.189	0.065	0.219	0.047	0.396	−0.013	0.797	0.008	0.881	−0.013	0.811	0.356	0.018 *	0.036	0.503
	C17:0%	−0.033	0.542	−0.025	0.663	0.022	0.695	−0.012	0.819	0.037	0.494	0.046	0.398	0.041	0.457	−0.357	0.012 *
	C20:0%	0.004	0.930	−0.030	0.576	−1.246	0.014 *	−0.015	0.771	1.670	0.034 *	0.043	0.426	0.017	0.755	−0.028	0.605
	C20:1%	1.733	0.043 *	0.643	0.026 *	0.050	0.358	0.070	0.189	0.095	0.082	0.035	0.518	0.096	0.077	−0.023	0.692
	C22:1n-9%	−0.067	0.177	−0.032	0.554	−4.441	0.006 **	−5.091	0.048 *	−5.553	0.028 *	−0.034	0.526	0.000	0.993	0.021	0.688
	C20:4n-6%	−0.007	0.882	−0.017	0.750	−0.027	0.629	−0.061	0.401	−0.035	0.527	0.027	0.625	−0.101	0.069	−0.016	0.038 *
	C18:3n-3%	1.370	0.035 *	−0.031	0.560	0.222	0.040 *	0.052	0.318	0.075	0.173	0.096	0.072	0.082	0.135	0.052	0.328
	C20:3n-3%	−0.018	0.712	0.004	0.938	−0.009	0.875	−0.111	0.008 **	−0.020	0.719	−0.020	0.706	0.008	0.878	0.007	0.903
	C20:5n-3%	0.031	0.536	−0.057	0.284	0.041	0.448	3.881	0.023 *	0.094	0.090	0.010	0.860	−0.062	0.259	0.064	0.232
	n-3 PUFAs%	0.005	0.916	0.034	0.522	−0.006	0.906	0.072	0.042 *	0.001	0.989	−0.003	0.960	−0.012	0.827	0.003	0.953

Multiple linear regression adjusted for age, gender, BMI, waist–hip ratio, culture, smoking, drinking, exercise, history of hypertension, diabetes mellitus, hypertriglyceridemia, and energy intakes. NW: normal weight group; OW: overweight group; OB: obese group; PUFAs: polyunsaturated fatty acids; MoCA: Montreal Cognitive Assessment; * *p* < 0.05; ** *p* < 0.01.

## Data Availability

The datasets used to support this study were not freely available in view of the participants’ privacy protection.

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
