# Peer review of "Association between the Erythrocyte Membrane Fatty Acid Profile and Cognitive Function in the Overweight and Obese Population Aged from 45 to 75 Years Old"

_nutrients, 2022, doi:10.3390/nu14040914_

Round 1
Reviewer 1 Report
- Under the first paragraph, please provide any information on the current obese/obesity rate among the elderly in China if possible.
- What makes the cut-off age of 45-75 in this study?
- In the background, list any previous studies that associated the erythrocyte fatty acids profiles with cognitive function. Please also include this in the discussion with a comparison of results.
- Under 2.1, how the participants were recruited within the 2 months? Was it through a health fair?
- Under 2.3.3, provide more details about the FFQ. Is it a validated questionnaire? Was it testing for the past year or past month past 6 months diet? How the dietary FFQ was administered? Which database was used.
- Under Table 1. "Culture" is listed as the categorical variable to indicate education levels? please double-check.
- Under the discussion, please discuss other potential confounding factors that may influence cognitive functions.
- Under Table 1, there were equivalent numbers of people with diabetes under each group (normal weight, overweight and obese), how did this affect the results of the study? Did you happen to analyze the data based on the A1C levels?
- Please provide the Cronbach’s alpha for the cognitive function survey for the current sample.
- Why multiple comparison corrections were not used in multiple analyses in the manuscript?
Author Response
Thank you for your precious comments and advice. Those comments are all valuable and very helpful for revising and improving our paper, as well as the important guiding significance to our researches. We have studied comments carefully and have made correction which we hope meet with approval. Revised portion are marked in red in the paper. The main corrections in the paper and the responds to the reviewer’s comments are as flowing:
Response to the reviewer's comments:
Q1: Under the first paragraph, please provide any information on the current obese/obesity rate among the elderly in China if possible.
Thank you for your careful review. According to the reviewer’s comment, we have added a more detailed information regarding the current obese/obesity rate among the elderly in China in the first paragraph as follows:
A cross-sectional study from eight regional centres in REACTION (China's Risk Evaluation of cAncers in Chinese diabeTic Individuals, a lONgitudinal study) found more than 64.4% residents aged over 40 years were peripheral obesity and/or central obesity (page 1, line 37)[1]. The reference was also added in the section 1. Introduction.
Q2: What makes the cut-off age of 45-75 in this study?
Thank you for your comments. The reasons that 45-75 years old participants were included in our study are as follows:
On the one hand, we aimed to study of the relationship between fatty acids compositions of erythrocyte membranes and cognitive functions of the middle-aged and elderly, and middle-age and elderly was here defined as populations with a mean age of 45 years and older[2], and 76 and 90 years old was considered advanced age[3]. On the other hand, older age was associated with higher rate of cognitive impairment[4] and the prevalence of MCI increased with age[5]. The Chinese community-dwelling populations over 55 years old had a pooled prevalence of 12.2% [95% confidence interval (CI): 10.6, 14.2%] for mild cognitive impairment (MCI) and 10.9% [95% CI, 7.7, 15.4%] for amnesic MCI (aMCI), respectively[5]. The positive rate of cognitive impairment among rural elderly aged 65 years and older was 42.9% (95% CI, 40.1-45.6) in north China[4]. Our previous study also found among the 675 participants aged 35-64 years old, 84 (12.4%) had MCI[6]. So we selected those who were 45-75 years as the research objects, and hoped to provide valuable data for its in-depth study in the future.
Q3: In the background, list any previous studies that associated the erythrocyte fatty acids profiles with cognitive function. Please also include this in the discussion with a comparison of results.
Thank you for your suggestion. As suggested by reviewer, we have added the suggested content to the manuscript on page as follows:
Previous study pointed that lower PUFAs and higher SFAs proportions in erythrocyte membrane fatty acid composition perhaps predict cognitive impairment in the elderly in Chinese (page 2, line 67). And more discussion with a comparison of this finding was added in 4. Discussion.
Q4: Under 2.1, how the participants were recruited within the 2 months? Was it through a health fair?
We are very grateful to your comments for the manuscript. The participants were recruited by the were recruited through the local village committee who informed the those consenting participants.
Q5: Under 2.3.3, provide more details about the FFQ. Is it a validated questionnaire? Was it testing for the past year or past month past 6 months diet? How the dietary FFQ was administered? Which database was used.
We are grateful for the suggestion. To be more clearly and in accordance with the reviewer concerns, we have added a more detailed information about the FFQ. In our study, the FFQ was used to ascertain the average intake frequency of 34 food items (including whole grain, red meat, pork, beef, mutton, chicken, duck, offal, fish, legume and legume product, milk, eggs, fruit and vegetables, nuts, sugared beverages, cooking oil, salt, etc.) in the past year. This questionnaire was adapted from the questionnaire used for the Dietary Investigation of Chinese Residents, which was organized by the Chinese Nutrition Society (CNS) in 2010[7]. The reference was also added in the section 2.3.3. Dietary intake survey.
The frequency of eating (daily, weekly, monthly, and annual) of each food in the FFQ was recorded in an interview by trained investigators. According to the results, the daily consumption per person of each food was calculated. The nutrient intakes and cumulative average daily intakes of each person was estimated by the Chinese food composition tables (the sixth edition) database.
Q6: Under Table 1. "Culture" is listed as the categorical variable to indicate education levels? please double-check.
Thank you for underlining this deficiency. As suggested by reviewer, we have replaced the “categorical variable” with “discontinuous variable” in our manuscript and the reanalysis could be found in Table 1.
Q7: Under the discussion, please discuss other potential confounding factors that may influence cognitive functions.
We deeply appreciate the reviewer’s suggestion. According to the reviewer’s comment, we have provided more details to discuss other potential confounding factors that may influence cognitive functions in the discussion section. In the original manuscript version, we have discussed the effect of age, BMI, waist-hip ratio and smoking on cognitive functions. After modification, we added the effect of culture, hypertriglyceridemia, history of hypertension and diabetes mellitus on cognitive functions (page 13, line 349).
Q8: Under Table 1, there were equivalent numbers of people with diabetes under each group (normal weight, overweight and obese), how did this affect the results of the study? Did you happen to analyze the data based on the A1C levels?
We are grateful for the suggestion. However, the numbers of people with diabetes under each group (normal weight, overweight and obese) are 47, 72, and 73, respectively.
Q9: Please provide the Cronbach’s alpha for the cognitive function survey for the current sample.
We deeply appreciate the reviewer’s suggestion. We used SPSS 26.0 to conduct reliability analysis. The Cronbach's alphas of MMSE and MoCA scales were 0.673 and 0.668, respectively. Internal consistency for the scales were generally good. The Cronbach Alpha Reliability classification occurs as follows: Very low (α ≤ 0.30); Low (0.30<α ≤ 0.60); Moderate (0.60 <α ≤ 0.75); High (0.75 <α ≤ 0.90); Very high (α> 0.90)[8].
Q10: Why multiple comparison corrections were not used in multiple analyses in the manuscript?
Thank you for your precious advice. Based on this suggestion, the pairwise analyses for continuous variables were performed by Bonferroni’s multiple comparison test and reanalysis could be found in Table 1, 2, 3, and 4, which was reflected in the change of the footmark. In response to this change, we have re-described in the Results and Discussion sections.
Thank you for your careful review. We really appreciate your efforts in reviewing our manuscript during this unprecedented and challenging time. We wish good health to you, your family, and community. Your careful review has helped to make our study clearer and more comprehensive.
References
- Qin, S.; Wang, A.; Gu, S.; Wang, W.; Gao, Z.; Tang, X.; Yan, L.; Wan, Q.; Luo, Z.; Qin, G., et al. Association between obesity and urinary albumin-creatinine ratio in the middle-aged and elderly population of Southern and Northern China: a cross-sectional study. BMJ open 2021, 11, e040214.
- Bartels, S.L.; van Knippenberg, R.J.M.; Dassen, F.C.M.; Asaba, E.; Patomella, A.H.; Malinowsky, C.; Verhey, F.R.J.; de Vugt, M.E. A narrative synthesis systematic review of digital self-monitoring interventions for middle-aged and older adults. Internet interventions 2019, 18, 100283.
- Fernández-Cao, M.L.; Camilli-Trujillo, C.; Fernández-Escudero, L. PROJECTA: An Art-Based Tool in Trauma Treatment. Frontiers in psychology 2020, 11, 568948.
- Wang, J.; Xiao, L.D.; Wang, K.; Luo, Y.; Li, X. Cognitive Impairment and Associated Factors in Rural Elderly in North China. Journal of Alzheimer's disease : JAD 2020, 77, 1241-1253.
- Lu, Y.; Liu, C.; Yu, D.; Fawkes, S.; Ma, J.; Zhang, M.; Li, C. Prevalence of mild cognitive impairment in community-dwelling Chinese populations aged over 55 years: a meta-analysis and systematic review. BMC geriatrics 2021, 21, 10.
- Fan, R.; Zhao, L.; Ding, B.J.; Xiao, R.; Ma, W.W. The association of blood non-esterified fatty acid, saturated fatty acids, and polyunsaturated fatty acids levels with mild cognitive impairment in Chinese population aged 35-64 years: a cross-sectional study. Nutritional neuroscience 2021, 24, 148-160.
- Zhang, W.; Li, Q.; Shi, L.; Lu, K.; Shang, Q.; Yao, L.; Ye, G. [Investigation of dietary intake of cadmium in certain polluted area of south in China]. Wei sheng yan jiu = Journal of hygiene research 2009, 38, 552-554, 557.
- Gottems, L.B.D.; Carvalho, E.M.P.; Guilhem, D.; Pires, M. Good practices in normal childbirth: reliability analysis of an instrument by Cronbach's Alpha. Revista latino-americana de enfermagem 2018, 26, e3000.

Reviewer 2 Report
Manuscript ID: nutrients-1589357
Dear Authors,
This study aimed to associate dietary fatty acids intake with cognitive function inthe overweight and obese population, by using the erythrocyte membrane fatty acid profile. Research is very interesting as well as it has a scientific value.
The introduction provides a good, generalized background of the topic that quickly gives the reader appreciation of the scientific relevance and timeliness of the research theme. I think the findings of this study are sufficiently described in the context of the published literature. The conclusions are supported by appropriate evidence.
However, there are flaws of the manuscript that need to be fixed
Specific comments on the manuscript are as follows:
- Please complete in the text of manuscript a subsection no. 2.5 – Statistical Analyses
- Please complete in the subsections: 2.4; 2.5 missing references. It is very important.
- Why weren’t the results of erythrocyte membrane fatty acid composition expressed as mol %?
- Please assess the practical use of the obtained results in the subsection Conclusions.
Overall it is a well-written article with an important high application potential.
Author Response
Thank you for your precious comments and advice. Those comments are all valuable and very helpful for revising and improving our paper, as well as the important guiding significance to our researches. We have studied comments carefully and have made correction which we hope meet with approval. Revised portion are marked in red in the paper. The main corrections in the paper and the responds to the reviewer’s comments are as flowing:
Response to the reviewer's comments:
Q1: Please complete in the text of manuscript a subsection no. 2.5 Statistical Analyses
We appreciate the reviewer’s positive evaluation of our work. Actually, the subsection has been completed in our manuscript. 2.6. Statistical Analysis
Q2: Please complete in the subsections: 2.4; 2.5 missing references. It is very important.
Thank you for your careful review. These two important pieces of research found in The association of blood non-esterified fatty acid, saturated fatty acids, and polyunsaturated fatty acids levels with mild cognitive impairment in Chinese population aged 35–64 years: a cross-sectional study and Erythrocyte membrane fatty acid profile & serum cytokine levels in patients with non-alcoholic fatty liver disease, respectively. We cited these two articles in the subsections: 2.4; 2.5.
Q3: Why weren’t the results of erythrocyte membrane fatty acid composition expressed as mol %?
Thank you for your comment. Because the using of internal standard led to the erroneous data and misinterpretation of experimental results, so fatty acids were expressed as the percentage of fatty acids curve area from the total fatty acids curve areas in the chromatogram and expressed as % instead of mol %.
Q4: Please assess the practical use of the obtained results in the subsection Conclusions.
We are grateful for the suggestion. To be clearer and in accordance with the reviewer concerns, we have added a brief description as follows in the subsection Conclusions: obese people, more precise dietary advice was needed to prevent cognitive impairment. Available data was provided for preventing obesity and cognitive decline by intaking dietary fatty acids plausibly. And the findings of our study provided relevant practical implications to the field for both researchers and practitioners, such as provide more accurate and individual food-based dietary guidance for overweight and obese people (page 16, line 491).
Thank you for your careful review. We really appreciate your efforts in reviewing our manuscript during this unprecedented and challenging time. We wish good health to you, your family, and community. Your careful review has helped to make our study clearer and more comprehensive.
